# Does an innovative paper-based health information system (PHISICC) improve data quality and use in primary healthcare? Protocol of a multicountry, cluster randomised controlled trial in sub-Saharan African rural settings

Xavier Bosch-Capblanch [1,2] Angela Oyo-Ita,[3] Artur Manuel Muloliwa,[4] Richard B Yapi,[5] Christian Auer,[1,2] Mamadou Samba,[6] Suzanne Gajewski,[1,2] Amanda Ross,[1,2] L Kendall Krause,[7] Nnette Ekpenyong,[3] Ogonna Nwankwo,[3] Anthonia Ngozi Njepuome,[8] Sofia Mandjate Lee,[9] Jahit Sacarlal,[10] Tavares Madede,[10] Salimata Berté,[5] Graça Matsinhe,[11] Abdullahi Bulama Garba,[12] David W Brown[13]

► Prepublication history and online supplemental material for this paper is available online. To view these files, please visit the journal online (http://dx.doi. org/10.1136/bmjopen-2021- 051823).

For numbered affiliations see end of article.

**Correspondence to**
Dr Xavier Bosch-Capblanch;
x.bosch@unibas.ch

## ABSTRACT

**Introduction** Front-line health workers in remote health facilities are the first contact of the formal health sector and are confronted with life-saving decisions. Health information systems (HIS) support the collection and use of health related data. However, HIS focus on reporting and are unfit to support decisions. Since data tools are paper-based in most primary healthcare settings, we have produced an innovative Paper-based Health Information System in Comprehensive Care (PHISICC) using a human-centred design approach. We are carrying out a cluster randomised controlled trial in three African countries to assess the effects of PHISICC compared with the current systems.

**Methods and analysis** Study areas are in rural zones of Côte d'Ivoire, Mozambique and Nigeria. Seventy health facilities in each country have been randomly allocated to using PHISICC tools or to continuing to use the regular HIS tools. We have randomly selected households in the catchment areas of each health facility to collect outcomes' data (household surveys have been carried out in two of the three countries and the end-line data collection is planned for mid-2021). Primary outcomes include data quality and use, coverage of health services and health workers satisfaction; secondary outcomes are additional data quality and use parameters, childhood mortality and additional health workers and clients experience with the system. Just prior to the implementation of the trial, we had to relocate the study site in Mozambique due to unforeseen logistical issues. The effects of the intervention will be estimated using regression models and accounting for clustering using random effects.

**Ethics and dissemination** Ethics committees in Côte d'Ivoire, Mozambique and Nigeria approved the trials.

## Strengths and limitations of this study

► These trials assess the effects of improving paper-based health information systems, which are greatly used particularly in remote, rural areas but which are neglected in research.

► The paper-based intervention (PHISICC) has been developed using a human-centred design approach, with front-line health workers and designers driving the cocreation process.

► Despite the complexity of health systems interventions, we have applied robust experimental methods, together with qualitative research, to assess and understand the effects of the paper-based intervention. Robust evidence on health systems is more likely to gain the credibility of policy-makers and to make it into systematic reviews, guidance development and policy and practice.

► Research targeting front-line health workers in remote, rural areas has to take place where they live and work, which poses serious obstacles in the organisation, management and monitoring of the trials.

► These obstacles, aggravated by the COVID-19 pandemic, have challenged the mobility of the research team, the availability of the intervention in one of the countries and the duration of the trials.

We plan to disseminate our findings, data and research materials among researchers and policy-makers. We aim at having our findings included in systematic reviews on health systems interventions and future guidance development on HIS.

**Trial registration number** PACTR201904664660639; Pre-results.

## INTRODUCTION

Front-line health workers (HW) in remote, rural health facilities (HF) in many countries are the first contact with the formal health sector of the population and they are confronted with life-saving clinical and public health decisions on a daily basis. Decisions are made by exerting a balanced judgement on the information related to healthcare events, such as making the correct diagnoses or deciding on which vaccinations a child should receive on a given day. In order to properly handle this information, appropriate data support tools and processes are required, referred to as the health information system (HIS), or routine HIS, or health management information system.[1] In reality, though, HISs are primarily designed to report aggregated health events to the higher tiers of the health systems rather than to inform decision making at the point of care.[2]

Increasing pressure by donors and governments to collect more and more data has aggravated the situation, through the proliferation of data support tools that have overloaded front-line HWs compromising their capacity to deliver good quality of care and to produce good quality data[3] for higher level decision making.

Promising 'quick fixes', such as the scale up of digital HIS, are taking a long time to implement and face enormous challenges related to infrastructure, equipment and services necessary to run them. Besides, research evidence on the effects of digital solutions remains patchy and inconsistent, even in high-income country settings, where complaints about computerisation of clinical care have been raised,[4,5]. Hence, it is very likely that paper tools will remain a primary, if not unique, data support mechanism, particularly in remote, rural HF in many countries.

Paper-based Health Information System in Comprehensive Care (PHISICC) is a multiyear, multicountry, mixed-methods research project that aims at producing and testing an innovative paper-based HIS to improve data quality and use, decision making and health outcomes, at primary healthcare (PHC). It is being carried out in selected areas within Côte d'Ivoire, Mozambique and Nigeria. The project started in 2015, producing a systematic review on the effects of HIS interventions[6 7] and a framework synthesis on how HIS are understood in the literature in order to learn from past experiences in HIS developments. This global evidence was coupled with studies to characterise the existing HIS in the three countries, to understand how HWs interact with the HIS and to identify entry points for HIS design improvements. With these bodies of evidence the research team was well equipped to engage into a human-centred design (HCD) cocreative process with front-line HW to design an innovative HIS (PHISICC). See figure 1 for an illustration of the structure, processes and evidence flow within PHISICC.

The impact of the PHISICC HIS on data quality and use, quality of healthcare and HW perceptions is being assessed concurrently in rural areas in the three countries. We describe the design of the trial here, consistent with Consolidated Standards of Reporting Trials reporting guidelines[8] and the extension for cluster randomised controlled trials (CRCT);[9] see online supplemental file 1.

## METHODS

### Aim

The aim of the trial is to address the research question: what are the effects of an innovative paper-based HIS (PHISICC) on data use and quality, quality of health and HW perceptions compared with the current HIS, in rural PHC settings?

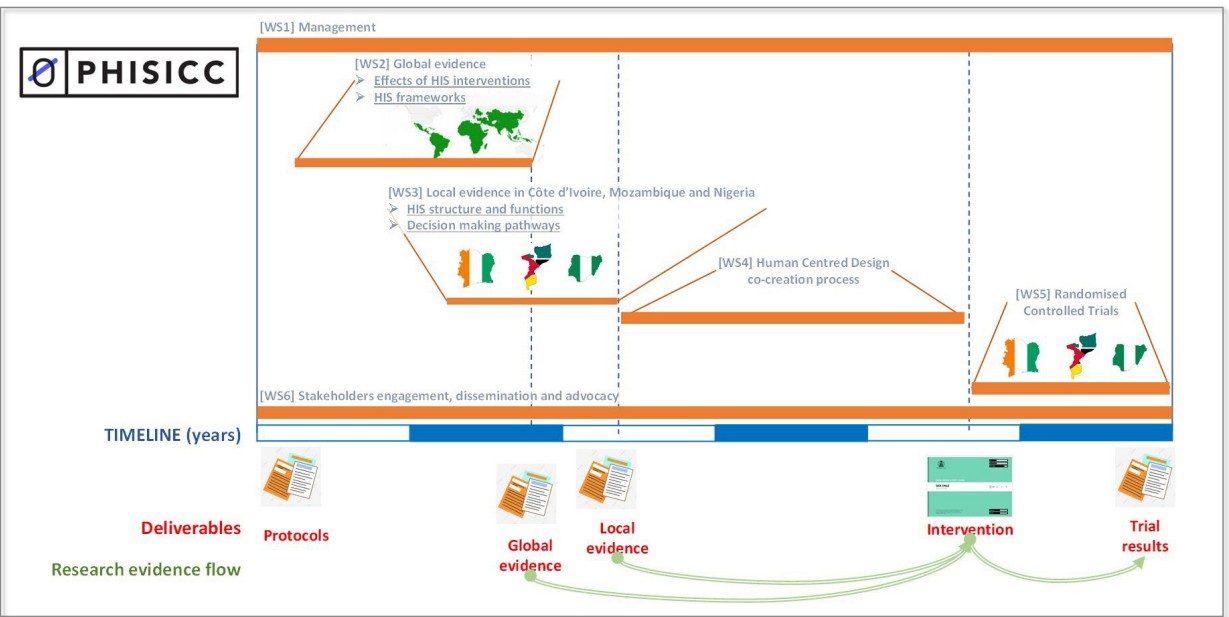

**Figure 1** PHISICC research programme structure, processes, deliverables and flow of evidence. Timelines are approximate. HIS, Health Information System; PHISICC; Paper-based Health Information System in Comprehensive Care; WS, work stream.

## Patient and public involvement

There was no public or patient involvement in the design of the study or selection of study areas because the intervention being assessed in these trials target healthcare providers and decision-makers, rather than patients or the public in general. Population in the catchment area of selected HFs, potentially using their services, were only approached in order to assess the studies outcomes.

On the other hand, we have involved health systems stakeholders and front-line HWs. Ministries of Health (MOH) at several levels participated in the preparation of the research proposal (personal consultations), in the characterisation of HIS that preceded the trials (countries workshops), and throughout all project components (additional workshops, newsletters and personal communications). Front-line HWs in the three countries have cocreated the intervention (ie, paper-based tools) through workshops, personal feedback and piloting under real live conditions. Some of them are part of the research team and coauthoring this manuscript.

## Study design

The study is a CRCT in each of the three countries. In each setting, 70 HFs are randomised to intervention or control (35 per arm). The intervention arm HF use the new PHISICC tools (substituting the usual HIS tools) and the control arm HF use the regular HIS tools. The trial is implemented in the real life contexts of HF carrying out their usual duties.

The trials started between the end of 2019 and beginning of 2020, depending on the country, when the intervention was installed and the baseline surveys carried out. Data collection will last until mid-2021.

## Study areas

MOH officials in several countries were contacted before submitting the proposal to the funding agency in order to explore the willingness to engage in a project focusing on paper-based tools. Officials in several countries rejected the offer on the grounds of upcoming digitalisation plans of the HIS in the country. We partnered with MOH that found the research relevant to their context in three countries.

In each country, the eligibility criteria of study areas were that they had to belong to the operational area of research partners; contain a large enough number of HFs and their catchment population; include vulnerable population (eg, with low vaccination coverage, high childhood mortality); and be comparatively neglected in terms of infrastructure and services. We excluded areas with concurrent research or other types of activities that could conflict with the CRCT (such as the coexistence of another health-related study, massive developments in infrastructure or activities involving migration of the population, such as temporary work sites or changes in working sites) and areas with threats to safety or security that could jeopardise research activities.

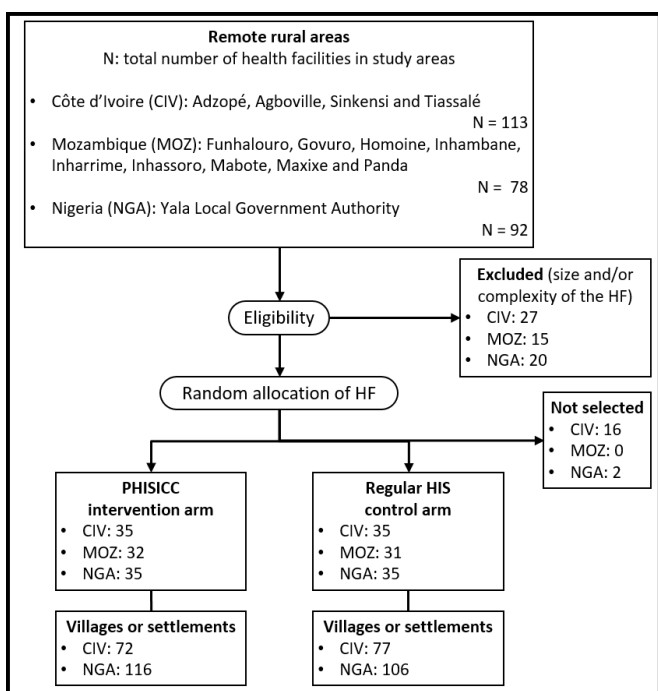

**Figure 2** CONSORT diagram: trial flow chart. CONSORT, Consolidated Standards of Reporting Trials; HF, health facilities.

The study areas are located in Adzopé, Agboville, Tiassalé and Sikensi districts (Côte d'Ivoire); in Funhalouro, Govuro, Homoine, Inhambane, Inharrime, Inhassoro, Mabote, Maxixe and Panda (Inhambane province, Mozambique) and in Yala Local Government Authority (Cross River State, Nigeria).

## Eligibility of HFs

The intervention is implemented at the HF level. The eligibility criteria of the HF were that they had to be located in the study areas, belong to the governmental health sector and their main activity should be the delivery of PHC services. HFs were excluded if they had specialised clinical services, inpatients, physicians providing care or with plans for staff turn-over.

A 'master list' of eligible HFs was prepared based on information provided by the MOH across all study areas. We aimed at selecting 70 of the eligible HF in each country, using simple random techniques in R.[10] See in figure 2 the selection and allocation trial flow chart.

## Allocation and blinding

Allocation of HF into the intervention and control arms took place in a formal event, gathering research partners and MOH officials to offer transparency and promote study ownership by local and national authorities. Equally sized, folded pieces of paper with the names and codes of included HF written on them were introduced in an opaque receptacle where they were manually and blindly mixed. A second receptacle contained two equally sized

pieces of paper, one with the word 'intervention' and another one with the word 'control'. A selected person in the meeting, not belonging to the research team, extracted one piece of paper at a time to reach half the number of included HF. Then, a paper was extracted from the second receptacle to assign those HF to the intervention or control arms. The rest of the papers were extracted as well to verify completeness and no duplication of names, and those HF were assigned to the other arm.

Once HFs were selected, all villages or settlements for each health facility catchment area were listed and three in each catchment area were selected. In practice, we selected all villages because the numbers were below (in Côte d'Ivoire) or just above (in Nigeria) the needs. For each village, we used Google satellite maps to identify and geo-locate every visible roof. Where there were many houses per village (roughly, more than fifty or so), a researcher would mark four points in the map slightly beyond the northernmost, southernmost, easternmost and westernmost roofs seen and 30 random points were selected within that square. From the mapped points, 10 per village (with 20 more acting as reserve) were randomly selected and marked on another map used in the field for data collectors to approach households. Where technical problems impeded this approach in a given village, a field supervisor would rotate a bottle on the floor towards the centre of the village and would select at random 10 households in the direction pointed by the bottle, from the outer limit of the village till the centre.[11]

Blinding is only feasible for the research team members carrying out the CRCT data collection and the analyses of the CRCT findings. The intervention (ie, paper tools) are by design very different from the existing system and it is not possible to blind participants or principal researchers.

We already had the agreement of the MOH and selected HF compliant with the inclusion criteria were provided with the intervention shortly after completing the baseline data collection. Therefore, recruitment as such took place at the same time of the allocation of HF into intervention and control arms.

## The intervention

The PHISICC paper-based intervention is a full set of paper-based tools to support decision making by frontline HW. These are the only tools to be used by HW in the intervention arm. The PHISICC tools encompass the whole system (ie, recording and reporting) and all clinical and public healthcare areas and are characterised by a common visual language (eg, spaces for digits and text), and standardised formats across healthcare areas. To support front-line HWs decision making, the PHISICC tools incorporate specific places to explicitly record critical data items (eg, respiratory rate in infants), graphic artefacts to distinguish severity degrees of signs or symptoms (ie, small square for 'normality', diamond for 'attention' and bold diamond for 'critical severity'); space to

document diagnoses and treatment decisions; and aides memories in the first page of register books.

The PHISICC tools have been developed from May 2017 till June 2019, including production, using a HCD approach.[12] A strength of the HCD approach is its ability to unlock the user's perspective so that designers can build solutions that are fully reality-based and work well. Cocreation groups were formed in each country with researchers, staff from partner institutions and healthcare workers, led by a team of professional designers. Research team members supervised and coordinated exclusively the feedback on the contents of the tools, to ensure compliance with each country clinical guidelines. At the outset of the process, the design focused on three healthcare areas (ie, antenatal care, vaccination and sick child) and slowly extended the new visual language to other healthcare areas. Initial workshops served to brainstorm on problems and potential design solutions, without any other rule than being comprehensive and not rejecting a single idea. Designers, then, formalised some of the most promising solutions and a first round of exchanges within the cocreation team was used to address misinterpretations or inconsistencies. There were two in-the-field testing rounds in Mozambique, two in Côte d'Ivoire and three in Nigeria and uncountable exchanges through teleconferences and email, in-between. The prototypes were considered final when no errors were detected, were compliant with data needs in each country and comments from the field could not be accommodated in the design concept or there was no consensus on minor or formal issues being raised.

The PHISICC tools have been produced in French for Côte d'Ivoire, in Portuguese for Mozambique and in English for Nigeria, which are the official languages used in the health systems in the three countries. They include the official logo of the MOHs. Healthcare areas covered include: family planning, antenatal care, including tetanus toxoid vaccination, delivery, postnatal care, vaccination, sick child, adults outpatient consultation, tuberculosis diagnosis and treatment, and HIV. Referral forms were also designed.

The PHISICC tools have three subcomponents: registers, tallies and reports. Registers are formed by seven Deutsches Institut für Normung (DIN) -A3 and one DIN-A4 (for referrals) book covering all healthcare areas except for tuberculosis treatment, for which DIN-A3 cards where used. Register books have 100, 200 or 400 pages depending on the country and healthcare area. They are used to record individual clients' data for each healthcare event, either of clinical or public health nature. Some register books have clinical notes at the very beginning, as 'aide memoires', and an example of a filled-in form, to assist HW when doubting how to proceed.

Tallies are DIN-A3 single sheets which contain a list of the indicators to be transferred to higher levels of the health system, with a series of small ovals, grouped in fives, to mark with tally sticks with a pen. In contrast to the current systems that have no tallies or only for

**Table 1** Comparison of new (intervention) and current (control) tools

| Characteristics | New (PHISICC intervention) tools | Old (control tools) |
|---|---|---|
| Development approach | Human-centred design, cocreation with users | Centrally done, based on data and information experts |
| Visual language | Standardised across tools | No visual elements |
| Information structure | Following clinical processes | Tabular form, following reporting requirements |
| Decision aids | Icons representing mild, moderate and severe conditions | Not available |
| Register books layout | Landscape, DIN-A3 | Depending on healthcare area; often much larger than DIN-A3 |
| Tally sheets to aid counting events | For each healthcare area, to be filled as healthcare events take place | Only for vaccination, to be filled as vaccinations take place |
| Reporting | Integrated with tallying/counting | Requires revisiting register books at the end of the month |

DIN, Deutsches Institut für Normung; PHISICC, Paper-based Health Information System in Comprehensive Care.

vaccination, tallies were created for all healthcare areas. In the middle-right side of the tally, a column accommodates cells aligned with the ovals to insert the count for each indicator; and in the far right of the sheet there is a replica of the count column, separated with a perforated line, which is detached and sent, as part of the monthly report to the higher level in the health system.

While current HIS tools are consistently organised in tabular formats and books, where each clinical event is recorded in a row and each variable (eg, age, gender, HIV status, diagnosis) in a column, PHISICC tools incorporated several innovations; in summary: a visual language to guide the clinical decisions of HWs based on severity (ie, if it is recorded that a child has convulsions, a visual artefact indicates severity), more space for clinical data (eg, vital signs), inclusion of all critical information to assess patients (eg, obstetric history, gestational age, fundus height, breath rate in infants), consolidation of information of all antenatal care visits for a single pregnant woman in the same page, among many other formal and contents improvements, including improved aesthetics. A systematic comparison of the new (intervention) and current (control) tools is provided in table 1.

We aimed at creating 'a system' (not just some tools) focusing on decision making by front-line HWs. The epidemiological and public health contexts in the three countries are similar, as confirmed by the similarities in the existing HIS between the three. The visual language and the recording forms where common to the three countries because clinical decisions are common to the three; although forms allowed for specific tests or treatments. The reporting component was adapted to each country set of indicators, although the visual language and reporting processes were harmonised.

During 3 or 4 days, HWs were trained on HIS before the start of the trial. In the intervention arm they were trained on the PHISICC tools; and the control arm received a refresher training about the regular tools, during the same number of days.

Additionally, given that the regular tools already contained information on past vaccination history of children still to complete their vaccination schedule, we created a mechanism to retrieve data of children's vaccination status to transcribe into the new vaccination register book in the intervention arm ('system transition').

Tools were endorsed by MOH, printed in local printing companies and distributed to HW at the end of the training sessions. A digital spreadsheet was created to monitor consumption and order additional tools to cover health facility needs during the life of the trial.

### Outcomes

There are five primary outcomes (table 2). Vaccination adherence is defined as the total number of vaccine doses given in the correct time interval to children in households in the HFs catchment villages over the total number of vaccine doses that should have been given during the same period. Antenatal care visits uptake will also be considered depending on the expected number of pregnancies in the study areas. Both are used as proxies for health outcomes in terms of protection against disease[13] and prevention of pregnancy complications[14] and are assessed in a random sample of households in the HFs catchment areas. Data concordance is defined as the level of agreement of HIS indicators between (1) records of healthcare events (recounts), (2) tallies (recounts) and (3) reports (aggregated data to higher levels of the system).[3] Decision making will be assessed considering the diagnostics scope in the sick child (ie, number of different diagnoses per sick child consultation) and treatment appropriateness (ie, number of prescribed treatments that are supported by a documented diagnosis). These outcomes will be assessed in a random sample of records and corresponding reports during the last 4 months of the study period. HWs satisfaction will be assessed in all HWs in included HFs using a standardised questionnaire.[15–17] While the intervention targets HF, some of the outcomes are measured at the level of HF and some others in the communities of HF catchment areas.

Secondary outcomes are classified under the following domains: data quality, data use, mortality, HW experience, client experience and resource consumption:

▶ Data quality, assessed in a sample of records

**Table 2** Outcomes and parameters used to estimate the sample sizes

| | Outcome name | Subjects | Definition | Baseline estimate | Expected change | Comments |
|---|---|---|---|---|---|---|
| 1 | Vaccination adherence | Children under 1 year (sample of households in catchment areas) | Number of vaccines given in the previous calendar year over the number of vaccine due in the same period | 75 given per 100 due | Increase of 10 per 100 | Vaccines are clustered within children, and children within HFs |
| 2 | Data concordance | Recording tools in health facilities (samples of records) | Number of healthcare events (eg, vaccinations, antenatal care consultations) recounted in the previous calendar year vs the number of healthcare events reported in the same time period | Seven recounted for each 10 reported[3] | Increase of 1 recounted | A single estimate can be obtained in each HF or by time periods (no clustering) |
| 3 | Diagnostic scope | Records of sick child consultations (samples of records) | Number of diagnosis in each sick child consultation during the previous calendar year | 30% with more than one diagnosis | 35% with more than one diagnosis | Individual consultations are clustered within HF |
| 4 | Treatment appropriateness | Records of sick child consultations (samples of records) | Number of treatments correctly prescribed in each sick child consultation during the previous calendar year | Half appropriate over all consultations | Increase to 60% | Individual consultations are clustered within HF (one treatment per child) |
| 5 | Health workforce satisfaction | Health workers (all health workers form include health facilities) | Degree (score) of satisfaction with the HIS across all health facilities in each arm | 50% satisfied | 75% satisfied | More than one health workers can be approached in each HF |

HF, health facility; HIS, Health Information System.

- – Completeness of recording and reporting in specific forms; that is, prevalence of unduly missing data items, partograph used.
  - – Accuracy of recorded figures in comparison to real events (eg, physical counting of commodities, such as number of 500 mg paracetamol tablets as recorded versus number of 500 mg paracetamol tablets as counted).
  - – Timeliness of reporting, as documented by time stamps in forms.
  - – Loss of data or data which does not reach the next upper administrative level.
- ► Data use, assessed in a sample of records
  - – In terms of knowledge (eg, vaccines due based on date of birth; weight for length assessments).
  - – Cases of different conditions properly treated (eg, diarrhoea cases given oral rehydration therapy according to national guidelines; pneumonia cases given appropriate antibiotic according to national guidelines).
  - – Public health decisions: availability of lost to follow-up lists or plans for vaccination, tuberculosis or HIV/AIDS treatment control.
  - – Occurrence of stock outs of essential drugs.
- ► Overall under-5s mortality and under-5s mortality excluding perinatal mortality,[18] in a sample of households in HFs catchment areas.
- ► HWs' 'human experience' and satisfaction (all HWs).
- ► District health information officers' 'human experience' (selected healthcare programme managers).
- ► Clients' 'human experience' and satisfaction, in a sample of households in HFs catchment areas.
- ► Resources consumption (eg, time use, costs)
  - – Intervention costs: tools, training, start-up.
  - – Time used for recording and reporting (eg, time-motion study).[19]
  - – Cost-effectiveness per unit of additional improvement in outcomes of interest.

It is worthwhile to note that outcomes that do not relate to data quality and use will be assessed using additional data collection tools (eg, survey questionnaires), which are the same for intervention and control HFs. Hence, the effects of the intervention cannot be attributed to the changes in performance of the paper tools routinely used to record healthcare events in intervention and control HFs, which are different by design.

In addition, we will consider 'explanatory outcomes' that will help to understand how the measured effects have taken place and why. We will look at the details of the interplay between the intervention, the system, the users and the context. Process indicators will be based on the documented activities that have taken place, from the conception of the intervention, up to its implementation, monitoring and evaluation. Process indicators may include: intervention setup and implementation, monitoring of the use of the intervention, special activities targeted at vulnerable populations, district reactions related to the intervention, handling of data coming from

the new system, sustainability based on costs information and perceptions, alignment with national health policies and donor priorities. We will also explore healthcare services characteristics looking at generic indicators from HFs, such as human resources profiles and relations with the communities, population characteristics and system and context characteristics captured in early stages of the project, where data are available.

## Sample size calculations

The required sample sizes for each primary outcome were determined using simulation to incorporate the clustering (table 1) and to take the baseline and end-line surveys into account. Briefly, we simulated 1000 trials with variation between them caused by drawing different samples from the same distributions. We then used the regression models detailed in the data section to analyse each of the simulated trials and estimate the power as the proportion of trials which detected the effect of the intervention as significant. The simulation code was written in R (online supplemental files 2 and 3).

For each country, we require the probability of $\alpha$, a type I error (rejecting the null hypothesis when it is actually true) to be less 0.05 and the power to be at least 80%.

For vaccination adherence, using a sample size of 35 HF per arm, we would have 80% power in each country to detect as significant a difference between a proportion of due vaccines given from 75% in the control to 85% in the intervention arms, assuming one child per household, 30 households per HF and a between-HF variation equivalent to a k of 0.25, where k is equal to the SD divided by the mean. The value of k is unknown, but was chosen in line with general observations by Hayes and Bennett.[20]

For data quality outcomes, with 35 HF per arm we would be able to detect as significant a difference from a ratio of 0.7 (reported:recorded) vaccinations in the control arm to 0.8 with the intervention with 80% power, assuming 100 recorded vaccinations per HF and an SD of 0.25 in the ratios between HF.

In terms of diagnostic scope, we would be able to detect an increase in the proportion of child visits with more than one diagnosis from 30% to 35% with 80% power with 35 HF per arm, 60 records per HF and assuming a k of 0.25.

We would be able to detect as significant an increase from 50% of treatments having a corresponding appropriate diagnosis to 60% with 80% power assuming 35 HF per arm, 1 treatment per child, 25 children per HF and variation between HF corresponding to k=0.25.

For the outcome related to HWs' satisfaction, we would be able to detect as significant an increase from 50% of HWs satisfied to 75%, with 80% power assuming 35 HF, three HWs per HF and a variation between HF equivalent to k=0.25. Since this variable is measured in the end-line survey only, we used the formula in Hayes and Bennett.[20]

In summary, in each country we require 35 HF per arm, 3 HW per HF, 100 vaccination records per HF, 60

sick child records per health facility and 30 children per health facility catchment area.

## Data collection and management

Data collection took place at baseline and will take place again at the end of the study. Data are collected from HFs, from the households in the catchment areas of the included HFs and also from district offices.

For data quality and data use outcomes, HF registers, tallies and reports will be scrutinised. For population-based outcomes, we carry out household surveys at baseline and at end-line. We use standard approaches for these types of surveys.[21] Households are visited, the research project is briefly introduced and consent requested. Ideally, mothers of alive children or women in childbearing age is interviewed in order to obtain information on living children (ie, vaccination history) and death events, respectively, using home-based records if available and accessible. Patients' satisfaction is assessed using the Patient Satisfaction Quaestionnaire (PSQ-18).[22 23 24] Essentially, the tool enables practitioners to investigate the extent to which their healthcare service meets the perceived needs of their client group and pinpoint areas for improvement.[24] The interview will be conducted with consenting individuals as close to their care encounter as possible.[25] Data tools are translated into the official languages of the study countries and pilot tested for consistent meaning and relevance to the setting. Data collectors are also able to communicate in local languages. The Satisfaction of Employees in Health Care survey is a validated tool to assess staff satisfaction. It was first developed and validated in a low-income country (Ethiopia)[26] and later successfully validated in a high-income country (USA).[27]

We use a mix of paper and electronic data (Open Data Kit (ODK)[28]) collection tools. Data collectors are trained to minimise error. Tools are piloted before implementing. ODK data are regularly stored and sent to secure servers, as soon as data collectors reach their office base. Data from paper tools are double entered and compared and sent to secure servers. Each data collection tool has its corresponding electronic database that is cleaned and submitted to the analyses. All data are anonymised at the point of data collection or as soon as possible in the data management process. Data are labelled with an arm code (eg, 'A' or 'B') without any further information allowing to disclose which data items belong to the intervention or to the control arms, ensuring blinding during data analyses.

Quality will be assured through several mechanisms: piloting of data collection tools; thorough training of field workers; checking missing data related; double, independent data entry from papers into digital databases; early descriptive analyses to detect potential outliers; field-workers tracking and supervision.

## Data analysis

The analysis will be carried out for each country separately, and based on intention to treat.

Baseline population and health facility characteristics (ie, basic demographic characteristics of population and HWs, professional profile of HWs, health facility size and services) will be summarised. If large imbalances are observed at baseline, the variables can be used to adjust the effect estimate comparisons.[29 30]

The analyses vary for the different primary outcomes due to the unit of measurement and levels of clustering, the type of variable, and whether measurements were taken at baseline and endpoint or endpoint only. We use regression models to allow us to estimate the effect of the outcome while flexibly accounting for these issues and allowing adjustment for potential confounders.

Logistic regression will be used for the binary variables: vaccine adherence is measured by determining whether each vaccine due was received, and treatment appropriateness by whether each treatment was correctly prescribed. Data concordance and diagnostic scope are count variables and may be analysed with Poisson regression, depending on their distribution. The regression model for HW satisfaction will depend on how it is distributed.

The outcomes have different levels of clustering (children or consultations, HW, HF). We will account for these levels of clustering by including random effects in the regression models.

Four of the primary outcomes are measured at baseline and end-line. The effect of the intervention will be estimated using an interaction term between arm and survey in the regression models: that is, is the change in the outcome between baseline and follow-up in the intervention arm different to the change between baseline and follow-up in the control arm? The effect of HW satisfaction in Nigeria, measured only at end-line, will be estimated as the difference between the intervention and control arms.

All estimates for the effect of the intervention will be presented with 95% CIs. The analyses will be carried out using R.[31]

## Measures to minimise bias

Statistical analyses will be carried out blindly, without knowledge of what HFs or population in the catchment area belong to the intervention or control arms. Only when the analysis code is considered as definitive and fixed, will results be shared with the wider investigators team and the arms for HFs and population will be disclosed.

Outcome measurement bias may take place where data from the HIS, which is the focus of the intervention, is used to measure outcomes. However, we will minimise this by assessing population based outcomes at household level.

Contamination (ie, the intervention affects individuals or units assigned to the control arm) may happen via the exchanges between HWs from HFs in both arms; for example, in monthly district data quality meetings, managerial meetings; or through inputs from supervisors

who influence control HFs with intervention tips encountered in HFs of the intervention arms. One mechanism to address this issue is using a district-based cluster randomisation scheme. However, we consider that (1) contamination, despite increasing the awareness of health works in control HFs, will hardly influence the decision making mechanisms that the HIS intervention focuses on and (2) randomisation at the level of district poses additional challenges that are not worth the marginal benefit of reducing a doubtful contamination.[32]

The spill-over effect (ie, benefits of the intervention extend beyond their direct recipients)[33] may take place in higher levels of the health systems; for example, district data managers and programme managers may experience the benefits of better structured and more timely data produced in HFs in the intervention arms. The trial will have no capacity to quantitatively account for spill overs at higher levels of the system, due to the limited number of higher level administrative areas that will be involved in the trial. However, through process indicators, we will consider potential benefits and harms of the intervention at higher levels of the system.

A challenge is the Hawthorne effect (ie, observer effect). Both HWs in the intervention and in the control sites will have an awareness of being observed as data collection activities will be at the same level of intensity in the two arms. Therefore, there should be no differential effect.

Analyses will be based on the intention to treat. It is important to closely monitor if the intervention HFs consistently use the new HIS tools and approaches. The data collection team and the trial monitoring team will check if old forms are still being used in the intervention HFs. However, we do not expect HFs to migrate between intervention and control arms, or vice versa, due to feasibility issues. On the other hand, some household members in a given catchment area may decide to seek for healthcare in a health facility belonging to another trial arm. In these cases, households will be analysed as belonging to the original trial arm.

## ETHICS AND DISSEMINATION
Ethics committees in Côte d'Ivoire, Mozambique and Nigeria approved the study in their respective countries. To date, some modifications to the protocol have taken place. In Côte d'Ivoire, we decided to select study areas close to the research institution based on logistics and practical reasons, instead of selecting an area in the north of the country, where poorer health indicators have been described. In Mozambique, the low density of HF per population implied extremely vast distances between HF and this, coupled with the rainy season, made the trial unfeasible in the originally selected Nampula province. After consultations, we decided to move the trial to the province of Inhambane and cancel the household survey. The allocation of HF to the intervention and control arms was completed using random number generation.

We plan to disseminate the findings of the trials as one of the few examples of studies assessing the effects of HISs interventions using experimental study designs.[34] Most of the experimental studies on HIS are circumscribed to specific healthcare areas (eg, tuberculosis, vaccination, cardiovascular disease) and very few have a system-wide approach (eg, PHC).[34] Experimental studies for health systems interventions are sometimes dismissed because of their limited capacity to provide reliable explanations of complex health system issues.[35] While we acknowledge these limitations, there is also a need for more robust evidence on the effects of these types of health system interventions[36] and there are also good examples of experimental studies reporting findings that can make it to the policy arena.[37] When embarking on this research, we considered from the outset the type of evidence required to be disseminated and included into systematic reviews,[38] guidance development[39] and eventually recommendations for policy and practice.[40]

We acknowledge the challenges of carrying out research on healthcare provided to remote, rural communities (in this case in sub-Saharan Africa). However, it is only in these remote areas where research about their specific problems and needs can take place. Challenges included long distances, poor conditions of roads, unreliable communications and limited food and accommodation services, all of them to be proactively handled to keep the quality of work and the morale of researchers and collaborators. We expect that the dissemination of findings in meetings, conferences and publications will contribute to a better understanding of what it takes to make research in challenging contexts.

The engagement and ownership of partners within this research has also been instrumental in order to plan and implement the CRCT. The intervention actually targets a governmental subsystem (the HIS) for which we required more than permission but also endorsement and active support. We achieved this level of collaboration by ensuring the participation of key stakeholders in key phases of the whole project, from inception till the implementation of the last phases, through frequent communications and workshops. The PHISICC programme includes targeted activities to keep decision-makers engaged and we are planning to share the findings through workshops as well as online and face-to-face events to disseminate the lessons learnt from the trial and the whole PHISICC research programme.

We also expect that the dissemination of our findings among partners and competitors will contribute to the current debates on the digitalisation of HISs. WHO recommendations on the matter are weak since the underlying evidence to support these recommendations is inconsistent.[41] Interestingly, the principles and methodological approaches in PHISICC can be applied to the development of any technological solution, being on paper, digital or mixed.

Finally, we expect that the results of the trials, both quantitative and qualitative, will be able to inform policies

on how to make HIS responsive to providers' decision-making needs, particularly in health services where the most vulnerable live.

## Author affiliations

[1] Swiss Tropical and Public Health Institute, Basel, Basel-Stadt, Switzerland
[2] University of Basel, Basel, Basel-Stadt, Switzerland
[3] Department of Community Medicine, University of Calabar Teaching Hospital, Calabar, Cross River, Nigeria
[4] Lurio University Faculty of Health Sciences, Nampula, Nampula, Mozambique
[5] Centre Suisse de Recherches Scientifiques en Côte d'Ivoire, Abidjan, Lagunes, Côte d'Ivoire
[6] Ministère de la Santé et de l'Hygiène Publique, Abidjan, Lagunes, Côte d'Ivoire
[7] Bill & Melinda Gates Foundation, Seattle, Washington, USA
[8] Swiss Tropical and Public Health Institute, Abuja, Nigeria
[9] Swiss Tropical and Public Health Institute, Maputo, Mozambique
[10] Faculty of Medicine, Eduardo Mondlane University, Maputo, Mozambique
[11] Expanded Program on Immunization, Ministério da Saúde, Maputo, Mozambique
[12] Planning, Research and Statistics, National Primary Healthcare Development Agency, Abuja, Nigeria
[13] BCGI LLC / pivot-23.5°, Chapel Hill, North Carolina, USA

**Acknowledgements** Members of the PHISICC Technical Advisory Group, chaired by David Brown: Blanche Anya (WHO AFRO), Abdallah Bchir (former Gavi), Marta Gacic-Dobo (WHO HQ), Richard Greffé (AIGA), Pamela Mitula (WHO AFRO), Sandy Oliver (UCL Institute) and Chris Wolf (BMGF). Research collaborators: Momade Ali, Celso Belo, Bassirou Bonfoh, Lisa Diallo, John Ferreira, Bernard Guessanbi, Caitlin Jarrett, Inza Koné, Felix Malé, Kouadio M'bra, David O'Donnell, Damaris Rodríguez (Sonder Collective), Melanie Wendland and Meike Zuske (Swiss TPH). Stakeholders from Côte d'Ivoire, Mozambique and Nigeria, participating in the consultation processes.

**Contributors** XB-C, AO-I, AMM, RBY and CA prepared the proposal for the funding agency, conceived the study and produced the data collection tools. SG ensured the regulatory, ethical and trial monitoring components. AR developed the analytical approaches and made the sample size calculations. RBY, MS and SB adapted the protocol to the context of Côte d'Ivoire and managed the administrative and ethical approvals in the country; AO-I, NE, ON, ANN and ABG likewise in Nigeria; AMM, SML and GM, in Nampula province (Mozambique); JS and TM adapted the protocol and acquired ethical and administrative clearances for Inhambane province (Mozambique). DWB is chair of the PHISICC Technical Advisory Group (TAG) and has coordinated multiple formal and informal inputs. LKK and DWB have advised on the adequacy of the study protocol within the overall PHISICC proposal and TAG advices. All country teams participated in PHISICC workshops and ensured that the protocol was suitable to countries realities; developed data collection tools and training materials. They are responsible for the implementation of the trial in each country. XB-C drafted the first version of the manuscript. All authors commented on several versions of the manuscript.

**Funding** This study was funded by the Bill & Melinda Gates Foundation, grant number INV-010193/OPP1135947.

**Competing interests** None declared.

**Patient consent for publication** Not required.

**Ethics approval** Comité National Ethique des Sciences de la Vie et de la Santé (CNESVS), reference: 024-19/MSHP/CNESVS-kp (Côte d'Ivoire); Comité Institucional de Bioética para Saúde da Universidade Lúrio, reference: 16.2/Julho/CBISUL/19 (Mozambique); Secretary, Government of Cross River State of Nigeria, Ministry of Health, Calabar Health Research Ethics Committee, reference: CRS/MH/HREC/018/Vol. V1/151 (Nigeria); Ethikkommission Nordwest- und Zentralschweiz (EKNZ), reference: 2018-01059 (Switzerland).

**Provenance and peer review** Not commissioned; externally peer reviewed.

**ORCID iD**
Xavier Bosch-Capblanch http://orcid.org/0000-0002-4469-0395

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
