## [Reviewer comments · BMJ Open]

ARTICLE DETAILS

TITLE (PROVISIONAL)	Does an innovative paper-based health information system (PHISICC) improve data quality and use in primary health care? Protocol of a multi-country, cluster randomised controlled trial in Sub-Saharan African rural settings.
AUTHORS	Bosch-Capblanch, Xavier; Oyo-Ita, Angela; Muloliwa, Artur; Yapi, Richard; Auer, Christian; Samba, Mamadou; Gajewski, Suzanne; Ross, Amanda; Krause, L Kendall; Ekpenyong, Nnette; Nwankwo, Ogonna; Njepuome, Anthonia; Lee, Sofia; Sacarlal, Jahit; Madede, Tavares; Berté, Salimata; Matsinhe, Graça; Garba, Abdullahi; Brown, David

VERSION 1 – REVIEW

REVIEWER	Tseng, Yu-hwei University of the Witwatersrand, Centre for Health Policy, School of Public Health, Faculty of Health Sciences
REVIEW RETURNED	26-Apr-2021

GENERAL COMMENTS	The authors have identified an important issue in the area of primary health care. It is always important to have voices from LMICs. I commend the authors for their efforts to improve quality of care by introducing a human centered paper-based health information system. Introduction (1) The authors reported that the project started in 2015 and that (a) a systematic review and a framework synthesis have been produced, and (b) studies that characterized existing HIS in the 3 countries. Nowhere in the manuscript has the authors provided these contexts prior to the current CRCT, which is essential. (2) It will be useful if the authors can provide a figure encompassing the qualitative, quantitative, structure, process and outcome elements of the whole project. Methods (1) The authors first indicated no patient involvement in the research. However, in the data collection section, they included a patient's satisfaction assessment. Clarification is required. (2) The process of co-creation of the intervention among frontline health workers was little described. What happened during and after workshops, personal feedback and piloting under real living conditions? How much time was spent? Who were involved? How did the researchers and health workers arrive at the final version? (3) How different is the new intervention from the existing tool? It will be useful if the authors provide a summary of what has been added to the new tool to provide a strong rationale for the change.
---

	(4) How did the researchers treat the heterogeneity of the three countries in terms of intervention design, health systems, the status of health workers in the health system and the scope of their service, data analysis and interpretation?
--	---

REVIEWER	McConnachie, Alex University of Glasgow, Robertson Centre for Biostatistics
-----------------	--

REVIEW RETURNED	02-May-2021
-------------

GENERAL COMMENTS	This review considers the paper by Bosch-Capblanch and colleagues, describing the design of a multinational cluster randomised trial of a paper-based health information system. This review focuses mainly on the statistical elements of the paper. I thought the abstract was fine, but the methods section reads more like how the trial was intended to be carried out, rather than what has actually happened. The abstract could perhaps recognise that there have been some difficulties implementing the trial as originally planned. I think the description of the study outcomes could be better. For example, the vaccination outcome reads as if it applies to the entire population of each health facility, whereas it is based on a surveys of households at baseline and follow-up. This is a little clearer in Table 1, but could be clearer in the text. In terms of the outcomes themselves, are the authors confident that they can be measured equally well, and in the same way, in the intervention and control HFs? The paper describes the data collection teams as being blind to the randomisation, which is good, but will they be able to stay blind when they start collecting some of the outcomes? Could the intervention actually improve some aspects of data collection (e.g. mortality data) and thereby make the outcomes for intervention HFs appear worse? The sample size section was not very clear, but I recognise that it is a very difficult part of the paper to get right. Would it help if the R code used for simulations were to be made available in the supplementary materials? That way, at least someone could replicate what was done. The authors state aiming for a Type 1 error rate of 5%, but do not say whether this included adjustment for having five primary outcomes. Crudely speaking, each outcome would have to be analysed at 1% significance. Also, the authors choose a value for k in their sample size calculations of 0.1, with reference to Hayes and Bennet, but I could not find any recommendation in that paper to match this assumption. The best I could find was a general statement that values are often no more than 0.25, and rarely more than 0.5. Given these two points, I do wonder whether the study could be underpowered. Is there any baseline data available that could inform the level of clustering of outcomes?
--

VERSION 1 – AUTHOR RESPONSE

Reviewer: 1	
Introduction	
(1) The authors reported that the project started in 2015 and that (a) a systematic review and a framework synthesis have been produced, and (b) studies that characterized existing HIS in the 3 countries. Nowhere in the manuscript has the authors provided these contexts prior to the current CRCT, which is essential.	We understand that the reviewer asks for a better narrative relating these research components. We have rephrased.
(2) It will be useful if the authors can provide a figure encompassing the qualitative, quantitative, structure, process and outcome elements of the whole project.	See Figure 2.
Methods	
(1) The authors first indicated no patient involvement in the research. However, in the data collection section, they included a patient’s satisfaction assessment. Clarification is required.	Clarified in the section. Patients were not involved in the research. We did approach community members, though, in the assessment of the outcomes.
(2) The process of co-creation of the intervention among frontline health workers was little described. What happened during and after workshops, personal feedback and piloting under real living conditions? How much time was spent? Who were involved? How did the researchers and health workers arrive at the final version?	We are very glad to read this comment, because we were being very synthetic here due to space concerns. We have given a better explanation now in the subsection “Intervention”.
(3) How different is the new intervention from the existing tool? It will be useful if the authors provide a summary of what has been added to the new tool to provide a strong rationale for the change.	See comment just above.
(4) How did the researchers treat the heterogeneity of the three countries in terms of intervention design, health systems, the status of health workers in the health system and the scope of their service, data analysis and interpretation?	An explanation has been added into the text.
Reviewer: 2	
I thought the abstract was fine, but the methods section reads more like how the trial was intended to be carried out, rather than what has actually happened. The abstract could perhaps recognise that there have been some difficulties implementing the trial as originally planned.	We have tried to be more explicit by adding some statements and deleting some terms in order to respect the abstract words limit.
I think the description of the study outcomes could be better. For example, the vaccination outcome reads as if it applies to the entire population of each health facility, whereas it is based on a surveys of households at baseline and follow-up. This is a little clearer in Table 1, but could be clearer in the text.	We have added detail, both in the narrative and in Table 1.
In terms of the outcomes themselves, are the authors confident that they can be measured equally well, and in the same way, in the intervention and control HFs? The paper describes the data collection teams as being blind to	This is really a good point that we have really discussed internally a lot. Clarification added after the list of secondary outcomes.

the randomisation, which is good, but will they be able to stay blind when they start collecting some of the outcomes? Could the intervention actually improve some aspects of data collection (e.g. mortality data) and thereby make the outcomes for intervention HFs appear worse?	
The sample size section was not very clear, but I recognise that it is a very difficult part of the paper to get right. Would it help if the R code used for simulations were to be made available in the supplementary materials? That way, at least someone could replicate what was done.	We have edited the sample size section for clarity. The simulation code is included as supplementary information.
The authors state aiming for a Type 1 error rate of 5%, but do not say whether this included adjustment for having five primary outcomes. Crudely speaking, each outcome would have to be analysed at 1% significance.	We limited the study to a small number of primary outcomes in which we were interested a priori, and do not plan to adjust the type 1 error rate.
Also, the authors choose a value for k in their sample size calculations of 0.1, with reference to Hayes and Bennet, but I could not find any recommendation in that paper to match this assumption. The best I could find was a general statement that values are often no more than 0.25, and rarely more than 0.5.	Apologies, we should have written $k=0.25$ (we can reproduce the numbers with the code with $k=0.25$).
Given these two points, I do wonder whether the study could be underpowered. Is there any baseline data available that could inform the level of clustering of outcomes?	We have corrected the k value. We do not have data on the level of clustering since there is little information on health systems from the rural HFs in general. However, the areas are fairly homogenous.

VERSION 2 – REVIEW

REVIEWER	Tseng, Yu-hwei University of the Witwatersrand, Centre for Health Policy, School of Public Health, Faculty of Health Sciences
REVIEW RETURNED	02-Jun-2021

GENERAL COMMENTS	The authors have addressed most of the questions I raised in the first review by adding the flowchart of the whole project and description of frontline health workers participation, and providing information about the new elements in the tool. Two questions for the authors after their addition.  1. An important characteristic of the new tool is user's participation. The authors also emphasized the decision making aspect by frontline health workers in the design of the new tool. Can they elaborate how this is operationalized and measured? 2. Can the authors provide a systematic comparison of the old and new tools in order to highlight the value of your efforts?
--

REVIEWER	McConnachie, Alex University of Glasgow, Robertson Centre for Biostatistics
REVIEW RETURNED	02-Jun-2021

GENERAL COMMENTS	The authors have addressed all of my original points. I have no further comments to make.
---

VERSION 2 – AUTHOR RESPONSE

REVIEWER 1

The authors have addressed most of the questions I raised in the first review by adding the flowchart of the whole project and description of frontline health workers participation, and providing information about the new elements in the tool.

1. An important characteristic of the new tool is user's participation. The authors also emphasized the decision making aspect by frontline health workers in the design of the new tool. Can they elaborate how this is operationalized and measured?

RESPONSE: we have rephrased in the "Intervention" subsection (for the operationalisation) and in the "Outcomes" section (form measurements).

2. Can the authors provide a systematic comparison of the old and new tools in order to highlight the value of your efforts?

RESPONSE: we have added a table with this comparison, which is referenced in the subsection "Intervention".

REVIEWER 2

The authors have addressed all of my original points. I have no further comments to make.

RESPONSE: thanks for this.

VERSION 3 – REVIEW

REVIEWER	Tseng, Yu-hwei University of the Witwatersrand, Centre for Health Policy, School of Public Health, Faculty of Health Sciences
REVIEW RETURNED	03-Jul-2021

GENERAL COMMENTS	The authors have well addressed my comments.
--